# Predictive Value of Cardiovascular Health Score for Health Outcomes in Patients with PCI: Comparison between Life’s Simple 7 and Life’s Essential 8

**DOI:** 10.3390/ijerph20043084

**Published:** 2023-02-10

**Authors:** Xueqin Gao, Xinrui Ma, Ping Lin, Yini Wang, Zhenjuan Zhao, Rui Zhang, Bo Yu, Yanhua Hao

**Affiliations:** 1Department of Social Medicine, School of Health Management, Harbin Medical University, Harbin 150086, China; 2Department of Cardiology, The Second Affiliated Hospital of Harbin Medical University, Harbin 150086, China

**Keywords:** cardiovascular health, Life’s Essential 8, Life’s Simple 7, major adverse cardiac events, percutaneous coronary intervention

## Abstract

The American Heart Association recently published an updated algorithm for quantitative assessments of cardiovascular health (CVH) metrics, namely Life’s Essential 8 (LE8). This study aimed to compare the predictive value between Life’s Simple 7 (LS7) and LE8 and predict the likelihood of major adverse cardiac events (MACEs) in patients undergoing percutaneous coronary intervention (PCI) to determine the utility of the LE8 in predicting CVH outcomes. A total of 339 patients with acute coronary syndrome (ACS) who had undergone PCI were enrolled to assess the CVH scores using the LS7 and LE8. Multivariable Cox regression analysis was employed to evaluate the predictive value of the two different CVH scoring systems at 2 years for MACEs. Multivariable Cox regression analysis revealed that both the LS7 and LE8 scores were protective factors for MACEs (HR = 0.857, [95%CI: 0.78–0.94], HR = 0.964, [95%CI: 0.95–0.98]; *p* < 0.05, respectively). Receiver operator characteristic analysis indicated that the area under the curve (AUC) of LE8 was higher than that of LS7 (AUC: 0.662 vs. 0.615, *p* < 0.05). Lastly, in the LE8 score, diet, sleep health, serum glucose levels, nicotine exposure, and physical activity were found to be correlated with MACEs (HR = 0.985, 0.988, 0.993, 0.994, 0.994, respectively). Our study established that LE8 is a more reliable assessment system for CVH. This population-based prospective study reports that an unfavorable cardiovascular health profile is associated with MACEs. Future research is warranted to evaluate the effectiveness of optimizing diet, sleep health, serum glucose levels, nicotine exposure, and physical activity in reducing the risk of MACEs. In conclusion, our findings corroborated the predictive value of Life’s Essential 8 and provided further evidence for the association between CVH and the risk of MACEs.

## 1. Introduction

Acute coronary syndrome (ACS) results from a plaque-related acute thrombus, causing major adverse cardiac events (MACEs) [1]. Percutaneous coronary intervention (PCI) remains the primary interventional treatment for ACS [2,3]. However, approximately 15–46% of patients suffering from cardiovascular diseases develop MACEs following PCI [4,5,6]. As a worldwide health issue, cardiovascular health (CVH) has generated extensive concern among global scholars. To date, the association between CVH and MACEs is being elucidated [7,8]. A previous study including 17,099 PCI patients who were followed up for 3 years determined that an optimal cardiovascular health score was related to a lower risk of cardiovascular events [9]. Similarly, research showed that lower cardiovascular health scores had a higher risk of MACEs and cardiovascular outcomes [10]. In another instance, a cohort study on 1277 individuals demonstrated that CVH was associated with a better prognosis after myocardial infarction [11]. These indicate that maintaining an ideal CVH may play a crucial role in preventing future MACEs in PCI patients. In 2010, the American Heart Association’s Strategic Planning Task Force and Statistics Committee designed Life’s Simple 7 (LS7) metric for monitoring the American Heart Association’s (AHA) 2020 impact goal to improve CVH [12]. However, in developing relevant health strategies by LS7, extensive evidence has provided insights that LS7 cannot accurately define and quantify CVH. Indeed, an evaluation system more comprehensive than “Simple” is urgently needed. 

More recently, an AHA Presidential Advisory presented an updated and enhanced approach to measuring, monitoring, and modifying CVH, referred to as Life’s Essential 8 (LE8) [13]. Several of the original metrics have been included and redefined in LE8. The new approach added sleep health as an eighth metric to the formal definition of CVH, as well as updated the remaining metrics such as diet, serum lipids and glucose levels, BMI, and nicotine exposure. A study conducted in Sweden based on 6537 individuals indicated that non-high-density lipoprotein cholesterol (non-HDL-C) parameters may be easier to calculate and interpret in clinical practice for the early prediction of future cardiovascular diseases [14]. This update on the original score enabled the LE8 to outperform previous scoring systems. Likewise, earlier studies signaled that sleep disturbance contributes to a broad range of disorders, including cardiometabolic diseases. Thus, incorporating sleep as a CVH metric may enhance primordial and primary cardiovascular disease (CVD) prevention efforts at the population level [15,16]. Therefore, sleep metrics may further enhance the value of LE8 in predicting CVD events. 

In addition, each metric has a new scoring algorithm ranging from 0 to 100 points, allowing the generation of a new composite cardiovascular health score that similarly varies from 0 to 100 points. All of these changes emphasize the “essentiality” of the new score. This update from “simplicity” to “essentiality” indicates that the new scoring system may have a different performance in evaluating CVH and be more sensitive to interindividual differences and intraindividual change than the previous scoring system [13]. Given the observations from previous reports, there is an urgent need to investigate the efficacy of this novel method of estimating and monitoring CVH in predicting diverse health outcomes. 

To the best of our knowledge, this is the first study to investigate the correlation between LE8 score and the risk of MACEs. Furthermore, whether LE8 can better predict the risk of MACEs than LS7 remains unknown. Therefore, this study aims to explore the predictive value of the two indicators to determine the risk of MACEs in patients undergoing PCI and to validate the utility of the new CVH scoring system in predicting CVH outcomes.

## 2. Materials and Methods

### 2.1. Patients and Procedure

A total of 350 consecutive patients with ACS who have undergone PCI at the Second Affiliated Hospital of Harbin Medical University between May 2019 and October 2019 were enrolled in this study. The exclusion criteria were as follows: patients with a history of coronary artery bypass graft (*n* = 2) or PCI (*n* = 4) and patients missing clinical data (*n* = 5). The final analytical sample consisted of 339 participants. 

Clinical and demographic information (age, sex, personal income, marital status, education level, medications, number of lesions, number of stents, etc.) at baseline were collected from the electronic health recording database. Based on these, the cumulative scores of the LS7 and LE8 components were determined. Similarly, outcomes of MACEs were collected through telephone follow-up 24 months after discharge. This study was approved by the Ethics Committee of the Second Affiliated Hospital of Harbin Medical University. Moreover, written informed consent was signed by all patients before being enrolled in the database. All acquired data were kept confidential.

### 2.2. Life’s Simple 7

The Life’s Simple 7 score was collected face-to-face during the patient’s hospitalization. Since the introduction of LS7 in 2010, our electronic health recording system started focusing on related metrics associated with CVH and has continuously recorded information on patients. In this study, all data were acquired from the electronic health recording system. The components of LS7 included diet, physical activity (PA), history of cigarette smoking, body mass index (BMI), total cholesterol levels, fasting blood glucose levels, and blood pressure (BP). Each metric was classified as poor (0), intermediate (1), or ideal (2). The overall score indicating the CVH status of the unit of assessment (individuals or populations) could range from 0 (all metrics at poor levels) to 14 (all 7 metrics at ideal levels). 

### 2.3. Life’s Essential 8

The components of LE8 include sleep health, diet, PA, nicotine exposure, BMI, blood lipids, blood glucose, and BP. Each metric has a new scoring algorithm ranging from 0 to 100 points, allowing the generation of a new composite cardiovascular health score (the unweighted average of all components) that also varies from 0 to 100 points. An overall CVH score of 80 to 100 is considered high CVH; 50 to 79, moderate CVH; and 0 to 49 points, low CVH. Our electronic health recording system has a comprehensive collection of patients’ data (dietary patterns, nicotine exposure, PA, sleep health, etc.). Therefore, the data were used to calculate the new LE8 score.

### 2.4. Outcome Variables

The primary endpoints were MACEs, including angina pectoris recurrence, severe arrhythmia, nonfatal myocardial infarction, congestive heart failure, revascularization interventions (PCI, percutaneous balloon dilatation), or cardiac death. Specifically, MACEs were defined as the time of first occurrence of these outlined events. An independent clinical events committee was established to adjudicate all events reported up to 24 h. Survival time was calculated from the date of hospitalization until the date of MACE occurrence.

### 2.5. Statistical Analysis

All statistical analyses were performed using SPSS version 25.0 (IBM, New York, NY, USA) and MedCalc version 19. Descriptive statistics were used to analyze baseline characteristics and demographics. Continuous variables were described using mean and SD, and categorical variables using frequency and percentage. Associations between continuous variables in MACEs groups were determined using the independent 2-sample *t*-test as appropriate. *χ*^2^ test was used to examine the association between categorical variables as appropriate. Multivariate Cox regression analyses ([HR] per 1 category increase, 95% confidence intervals [CI]) were divided into two steps. The first step was to analyze the associations between MACEs and the two cardiovascular health scores using multivariate Cox regression analysis. During this process, the final multivariate Cox analysis was adjusted for potential confounding covariates that were statistically significant with MACEs in the univariate analysis (*p* < 0.05). A receiver operating characteristics (ROC) analysis was performed in this step. Area under the curve (AUC) analysis was employed to compare the predictive abilities of two scoring systems for MACEs. The second step was to identify the interaction between the 8 CVH metrics and MACEs in the multivariate Cox regression analysis and assess whether they were correlated with MACEs in ACS patients. The results are presented in the forest plots. The results of these analyses were expressed as hazard ratio (HR) and 95% confidence intervals. Metrics not significantly different in the multivariate Cox regression analysis were further inputted in interaction analysis with other significant metrics. Each interaction analysis was used to reflect whether three nonsignificant metrics contributed to predictive effect through interaction with other significant metrics. A *p*-value < 0.05 was considered statistically significant.

## 3. Results

### 3.1. Differences in Baseline Characteristics among the MACE and Non-MACE Groups of the Study Population

The demographics and clinical baseline characteristics of the patients are presented in Table 1. Out of 339 participants with incident ACS, 105 (31.0%) had MACEs (32 angina pectoris recurrence, four severe arrhythmias, four cardiac deaths, eight recurrent nonfatal myocardial infarctions, 45 revascularizations, and 12 congestive heart failure) during a follow-up period of 2 years. Among the participants, the mean age was 59.57 ± 11.18 years, and 246 (72.6%) patients were male. Compared with non-MACEs groups, the LE8 score was significantly lower in the MACEs group compared to the non-MACEs groups (51.21 ± 11.81, *p* < 0.001) in the MACEs group (Table 1). All other demographic characteristics were comparable between the two groups, with the exception of living patterns.

### 3.2. Predictive Value of Life’s Essential 8 and Life’s Simple 7

As listed in Table 2 (Model 1), multivariate Cox proportional hazards analysis for the prediction of MACEs was performed for patients’ baseline CVH scores. The results uncovered that the LS7 score was a predictor of a lower risk of MACEs after adjusting for the number of lesions, number of stents, and living patterns (HR = 0.857, [95%CI: 0.78–0.94], *p* < 0.05). In contrast, further analysis using the multivariate Cox proportional hazards analysis showed that the LE8 score was a significant influencing factor of MACEs risk after adjusting for the number of lesions, the number of stents, and living patterns (HR = 0.964, [95%CI: 0.95–0.98], *p* < 0.05) (Table 2, Model 2).

Figure 1 displays the results of the Area Under the Curve (AUC) analyses. According to the Receiver Operating Characteristics curve, it can be deduced that LS7 and LE8 had an excellent discriminative performance for differentiating MACEs, with AUCs of 0.615 and 0.662, respectively. Furthermore, the AUC for LE8 was higher than that of LS7, implying that the former score had higher predictive abilities than the latter (*p* < 0.05). 

Figure 2 illustrates the Multivariate Cox analysis results of each metrics’ predictive value for MACEs using LE8. Diet (HR= 0.985, [95%CI: 0.97–0.99]; *p* = 0.015), sleep health (HR = 0.988, [95%CI: 0.98–0.99]; *p* < 0.001), blood glucose levels (HR = 0.993, [95%CI: 0.98–0.99]; *p* = 0.039), nicotine exposure (HR= 0.994, [95%CI: 0.99–0.99]; *p* = 0.025) and PA (HR = 0.994, [95%CI: 0.99–0.99]; *p* = 0.025) were found to be predictors of MACEs. Contrastingly, health factors domains, blood lipid levels, BMI, and BP were not significantly associated with MACEs. Therefore, interaction analysis was conducted among blood lipid levels, BMI, BP, and the remaining five metrics and determined that the interaction effect between these three metrics and the others was significantly associated with MACEs (*p* < 0.05) (Table 3).

## 4. Discussion

To our knowledge, this is the first study to compare the predictive value between the LS7 and LE8 to determine MACEs. The present results demonstrated that LE8 outperformed the original scoring system in predicting MACEs. Furthermore, after analyzing each metric, diet, sleep health, blood glucose, nicotine exposure, and PA were found to be significantly associated with MACEs. Our study further demonstrates the predictive value of LE8 for MACEs from these metrics. 

Based on our findings, both LS7 and LE8 were associated with MACEs. Improving the CVH score may be an effective approach to lowering the risk of MACEs. Although the original LS7 was correlated with the risk of MACEs, according to the ROC curve of the two scores, our study suggested that LE8 had a better performance in predicting MACEs than LS7. This may be explained by the following reasons: Firstly, each component is further categorized into three categories in LS7: ideal, intermediate, and poor, whereas the new score allows each metric a new scoring range from 0 to 100 points, which has more coverage than the original score. In that case, it could be more challenging for individuals to exhibit healthy behaviors and achieve an ideal score than in the previous LS7 score. Moreover, individual variances in CVH were amplified, specifically in clinical and research settings. This may partially explain the difference in predictive value between the two scores in predicting MACEs. Furthermore, the LE8 writing group recommended incorporating sleep duration for capturing sleep health, secondhand smoke exposure, and multidimensional diet factors in Mediterranean Eating Pattern, which are known to have adverse effects on cardiovascular and overall health [13]. In addition, hemoglobin A1c and non-HDL-C levels were used to measure serum glucose and lipid levels rather than fasting blood glucose and total cholesterol, which better reflect lipids and glycemic control among diabetic patients [17,18]. In this study, the overall CVH scores using LE8 were significantly related to MACEs. Given the aforementioned findings, pursuing an overall ideal CVH may be an essential way to reduce the risk of MACEs in acute myocardial infarction patients in the future.

In this updated CVH score, the eight metrics making up the new CVH definition have been grouped into the two domains of health behaviors (diet, sleep, nicotine exposure, and PA) and health factors (BMI, blood lipids, blood glucose, and BP) [13]. Our results validated that all metrics in the health behaviors domain and blood glucose levels were significantly associated with MACEs (HR= 0.985, 0.988, 0.993, 0.994, and 0.994, respectively). Although in health factors domains, blood lipids, BMI, and BP were not independent predictors of MACEs, those three metrics remained significantly associated with MACEs after interaction analysis (*p* < 0.05). This interaction effect could enhance the prediction of MACEs. Considering the observations in the present study, all components of LE8 should be taken into account when evaluating and monitoring CVH.

The results of this study indicate that diet has the highest impact on the risk of MACEs. Unlike LS7, a modified Mediterranean Eating Pattern for Americans as the tool for measurement of diet was proposed for LE8 [13], which is regarded as a rapid dietary assessment tool for individuals. This new tool takes olive oil, berries, fast food, nuts, and so forth into account. According to our results, patients with a low diet score were more prone to develop incident MACEs (HR= 0.985, [95%CI: 0.97–0.99]; *p* = 0.015). The findings may be associated with the consumption of olive oil and berries in the patient’s daily diet. As is well documented, high fruit intake may offset the detrimental effect of lipid intake on hypertension [19]. In addition, berries serve as the primary dietary sources of anthocyanins. A previous study reported, high dietary anthocyanins are related to a lower risk of coronary heart disease (RR: 0.83, [95%CI: 0.72–0.95]; *p* = 0.009) [20]. However, fewer patients consumed adequate berries (≥2 servings of berries per week) in this study. Furthermore, stir-frying is one of the most common cooking methods in China. While most families use soybean oil, peanut oil, and lard for cooking, olive oil is rarely used. Prior studies have suggested that substituting soybean oil with olive oil is beneficial for cardiometabolic health [21,22]. We recommend prioritizing the use of olive oil instead of other vegetable oils in cooking. As a result, healthcare professionals should encourage PCI patients to adhere to healthier diets, promoting ideal overall cardiovascular health scores and thereby reducing the occurrence of MACEs.

Sleep is a foundational element of human biology and a requirement for life [23]. A previous study demonstrated that patients with multidimensional sleep health had a 47% lower incident CVD risk [16]. In the present study, poorer sleep health was associated with a 1.2% increased risk of MACEs. Likewise, results from more recent studies showed that sleep profile was significantly associated with an increased risk of MACEs [24]. It is worthwhile pointing out that sleep health is a critical component in CVH and may improve CVD risk prediction and enhance CVD primordial and primary prevention efforts. Therefore, it is imperative for patients at high risk of MACEs to pay more attention to their sleep health.

Secondhand tobacco smoke exposure has also been added to the definition of LE8 to reflect its adverse impact on health besides traditional combustible tobacco and inhaled nicotine delivery systems [13]. The range of nicotine exposure is more detailed in LE8, which can comprehensively reflect the nicotine exposure of patients. It is widely accepted that quitting smoking is associated with significant risk reduction in adverse outcomes among patients with CHD [25]. Our current study showed that nicotine exposure was associated with a 0.6% increased risk of MACEs (HR= 0.994, [95%CI: 0.99–0.99]; *p* = 0.025), which implied that nicotine exposure should be continuously monitored in patients with cardiovascular disease.

Studies have shown that PA is associated with cardiometabolic markers and may be a means of preventing cardiovascular disease. In addition, previous research results confirmed that physical activity, to some extent affects the patient’s cardiovascular health, and lack of adequate PA resulted in cardiovascular adverse events occurring more frequently [26,27]. In the present study, poor PA was associated with a 0.6% increased risk of MACEs (HR= 0.994, [95%CI: 0.99–0.99]; *p* = 0.025), which suggests that persistent moderate or greater intensity PA can effectively improve CVH, thus reducing the risk of MACEs. Our results are consistent with the findings of previous studies, in which 1–2 days per week of moderate PA or more were significantly associated with a lower risk of MACEs [28]. On this basis, our results add to the body of evidence to support the association between PA and the risk of MACEs. Additionally, despite the new LE8 not specifically pointing out the possible impact of vigorous PA, other studies described that MACEs were significantly more frequent in those undertaking competitive sports [29]. Therefore, it is necessary to pay attention to the intensity of PA in clinical practice. According to the updated and enhanced physical activity recommendations in LE8, vigorous and moderate intensity PA are defined as any activity used for sports, fitness, or recreational activities that results in a substantial increase in breathing or heart rate, such as running or basketball for at least 10 consecutive minutes; and activities that result in a small increase in breathing or heart rate, such as brisk walking, bicycling, swimming, or playing golf for at least 10 consecutive minutes (excluding work, transportation, or household chores) [30]. Moreover, according to the PA recommended in LE8, to ensure that PA can pursue 100 full marks, it should be guaranteed to proceed with 150 min of moderate- (or greater) intensity activity per week [13].

In health factor domains (BMI, blood lipids, blood glucose, BP) of LE8, our results showed that blood glucose levels were significantly associated with MACEs. It is interesting to note that, although these metrics (blood lipid levels, BMI, and BP) were not significant with MACEs in the multivariate analysis, they were able to correlate with MACEs after interaction analysis with other significant metrics (Diet, sleep, nicotine exposure, blood glucose, and PA). A previous study demonstrated an interaction between diet, BMI, BP, and other risk factors for CVD [31,32]. This observation is in line with the findings of this present study. As emphasized in LE8, maintaining the highest possible levels of CVH on all metrics leads to optimal outcomes. Our study again reiterates that focusing on the overall CVH could promote the predicted value of the risk of MACEs.

## 5. Conclusions

Both Life’s Simple 7 and Life’s Essential 8 showed robust value in predicting the risk of future MACEs. Compared to Life’s Simple 7, Life’s Essential 8 displayed a superior performance. The current analysis corroborated the predictive value of Life’s Essential 8 and provided further evidence for the association between CVH and the risk of MACEs.

## 6. Limitations

Our study has potential limitations that must be considered when interpreting the results. Firstly, this was a single-center study, and the sample size was small. Larger multi-center studies should be conducted to verify the effect on MACEs. In addition, our center mainly performs PCI treatment, and CABG patients were not included in this sample. Secondly, the follow-up period was only two years, and long-term dynamic follow-up may be considered in the future. Moreover, the patient sample in this study originated from a homogenous population (same country, presumably similar culture, eating habits, and lifestyle), and results may not be generalizable to other countries or cultures. Finally, the available data used were collected by the research group in previous studies, which have been extracted and converted from medico-administrative databases. Comprehensive data acquisition can be performed in the future to validate the results of this study.

## Figures and Tables

**Figure 1 ijerph-20-03084-f001:**
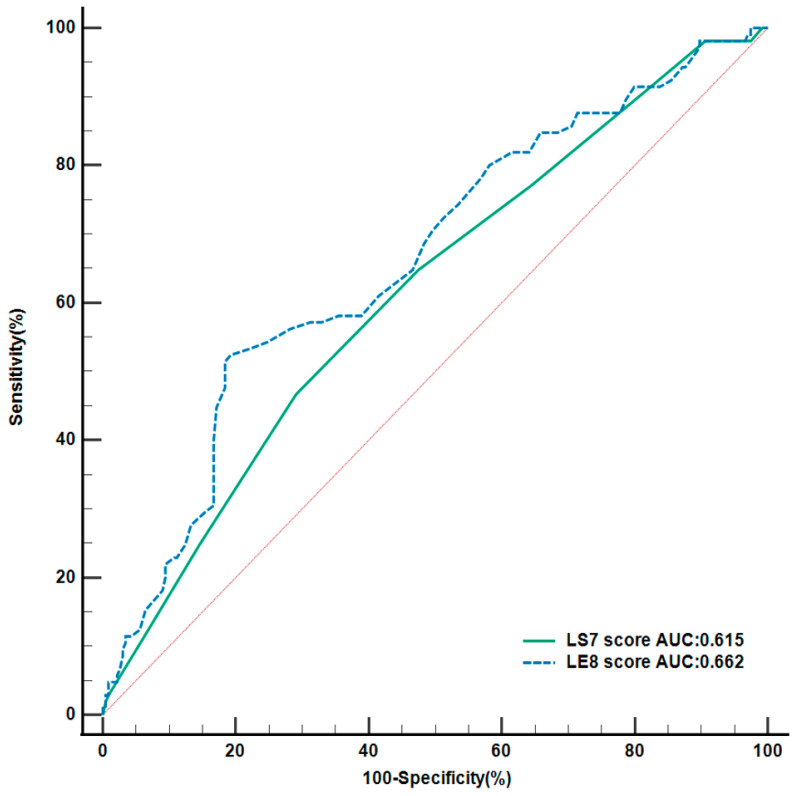
The ROC curve of Life’s Simple 7 and Life’s Essential 8 for predicting MACEs. LS7 Score, Life’s Simple 7 cardiovascular health score; LE8 Score, Life’s Essential 8 cardiovascular health score. 170 × 162 mm(×DPI).

**Figure 2 ijerph-20-03084-f002:**
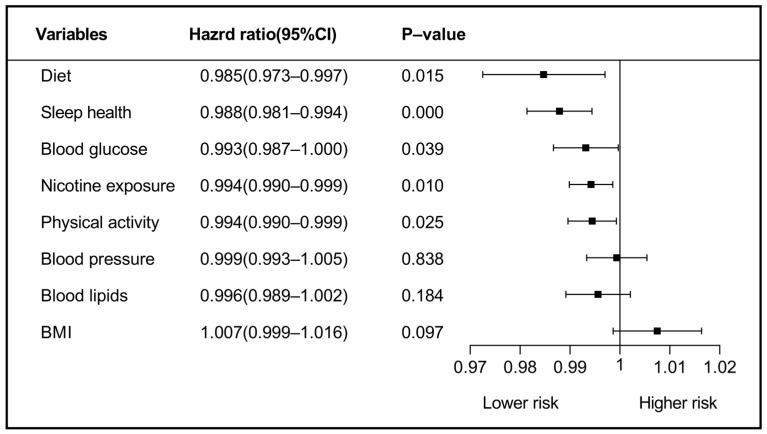
Cox regression of the 8 metrics of LE8 on the prediction of MACEs. BMI, body mass index. 174 × 99 mm(×DPI).

**Table 1 ijerph-20-03084-t001:** Patients Demographics and Baseline Clinical Characteristics (*n* = 339).

Variables	Total Population	MACEs (+)	MACEs (-)	*p* Value
	*n* = 339	*n* = 105	*n* = 234	
Age (y) ^†^	59.57 ± 11.18	59.42 ± 11.21	59.64 ± 11.19	0.869
Sex				0.959
Male	246 (72.6)	76 (72.4)	170 (72.6)	
Habitation				0.815
City	197 (58.1)	62 (59.0)	135 (57.7)	
Rural	142 (41.9)	43 (41.0)	99 (42.3)	
Living pattern				0.035
Alone	29 (8.6)	14 (13.3)	15 (6.4)	
Not-Alone	310 (91.4)	91 (86.7)	219 (93.6)	
Marital status				0.566
Have spouse	34 (10.0)	12 (11.4)	22 (9.4)	
No spouse	305 (90.0)	93 (88.6)	212 (90.6)	
Educational attainment				0.200
≤Junior high	195 (57.5)	55 (52.4)	140 (59.8)	
≥Senior high	144 (42.5)	50 (47.6)	94 (40.2)	
Personal income (RMB/month)				0.865
≤2500	172 (50.7)	54 (51.4)	118 (50.4)	
>2500	167 (49.3)	51 (48.6)	116 (49.6)	
Medications				
Dual Antithrombotic therapy				0.730
No	10 (2.9)	4 (3.8)	6 (2.6)	
Yes	329 (97.1)	101 (96.2)	228 (97.4)	
Statins				1.000
No	2 (0.6)	1 (1.0)	1 (0.4)	
Yes	337 (99.4)	104 (99.0)	233 (99.6)	
β-blockers				0.079
No	96 (28.3)	23 (21.9)	73 (31.2)	
Yes	243 (71.7)	82 (78.1)	161 (68.8)	
ACEI/ARB				0.795
No	165 (48.7)	50 (47.6)	115 (49.1)	
Yes	174 (51.3)	55 (52.4)	119 (50.9)	
Number of lesions ^†^	2.49 ± 0.82	2.65 ± 0.78	2.41 ± 0.83	0.015
Number of stents ^†^	1.38 ± 0.97	1.59 ± 1.17	1.28 ± 0.85	0.006
LS7 ^†^	6.42 ± 2.08	5.83 ± 2.01	6.68 ± 2.06	<0.001
LE8 ^†^	55.88 ± 12.12	51.21 ± 11.81	57.97 ± 11.69	<0.001

MACE, major adverse cardiac events; ACEI, Angiotensin-Converting Enzyme Inhibitors; ARB, angiotensin receptor blocker; LS7, Life’s Simple 7 cardiovascular health score; LE8, Life’s Essential 8 cardiovascular health score. ^†^ Data are means ± standard deviations.

**Table 2 ijerph-20-03084-t002:** Cox regression of the Life’s Simple 7 and Life’s Essential 8 on the prediction of MACEs in 2 years (*n* = 339).

Variables	Model 1			Model 2		
	HR	95%CI	*p* Value	HR	95%CI	*p* Value
Number of lesions	1.317	1.021–1.699	0.034	1.312	1.024–1.681	0.032
Number of stents	1.197	0.999–1.434	0.051	1.153	0.961–1.384	0.126
Living pattern (Not-Alone)	0.611	0.348–1.075	0.088	0.623	0.354–1.095	0.100
LS7 Score	0.857	0.780–0.943	0.001	-	-	-
LE8 Score	-	-	-	0.964	0.948–0.980	<0.001

Adjusted Cox regression analysis of associations between LS7, LE8 and MACEs. Model 1: Cox regression of the LS7 Score. Model 2: Cox regression of the LE8 Score. LS7 Score, Life’s Simple 7 cardiovascular health score; LE8 Score, Life’s Essential 8 cardiovascular health score.

**Table 3 ijerph-20-03084-t003:** Interaction analysis of Blood lipids, BMI and BP with other 5 metrics ^†^ in LE8 (*n* = 339).

Variables	HR	95%CI	*p* Value
Blood lipids			
Model 1: Diet ^a1^	0.9997	0.9996–0.9999	0.005
Model 2: Sleep health ^a2^	0.9998	0.9997–0.9999	<0.001
Model 3: Nicotine exposure ^a3^	0.9999	0.9998–0.9999	0.028
BMI			
Model 1: Diet ^b1^	0.9997	0.9995–0.9999	0.007
Model 2: Sleep health ^b2^	0.9998	0.9997–0.9999	<0.001
Model 3: Nicotine exposure ^b3^	0.9999	0.9998–0.9999	0.019
Model 4: Physical activity ^b4^	0.9999	0.9998–0.9999	0.047
BP			
Model 1: Blood glucose ^c1^	0.9998	0.9996–0.9999	0.009
Model 2: Sleep health ^c2^	0.9998	0.9996–0.9999	0.012
Model 3: Nicotine exposure ^c3^	0.9998	0.9997–0.9999	0.012

^†^ Cox regression analysis results in figure showed Blood lipids, BMI and BP were not significant with MACEs, we further conduct interaction analysis of these three metrics with Diet, Sleep health, Blood glucose, Nicotine exposure, and Physical activity. BMI, body mass index; BP, blood pressure. ^a1^, Blood lipids × Diet; ^a2^, Blood lipids × Sleep health; ^a3^, Blood lipids × Nicotine exposure. ^b1^, BMI × Diet; ^b2^, BMI × Sleep health; ^b3^, BMI × Nicotine exposure; ^b4^, BMI × Physical activity. ^c1^, BP × Blood glucose; ^c2^, BP × Sleep health; ^c3^, BP × Nicotine exposure.

## Data Availability

The data underlying this article are available in the article.

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
