# Peer review of "Predictive Value of Cardiovascular Health Score for Health Outcomes in Patients with PCI: Comparison between Life’s Simple 7 and Life’s Essential 8"

_ijerph, 2023, doi:10.3390/ijerph20043084_

Round 1

Reviewer 1 Report

This article by Gao et al aims to establish which one of two scoring systems better predicts major adverse cardiac events. The two scoring systems contain classical factors such as diet, serum lipids and glucose levels, sleep, BMI and nicotine exposure.

Title should not contain abbreviations

Abstract: please explain how scores are protective factors for a disease. Please explain why the area under the curve is important, what its significance is. Last sentece of the abstract brings no scientific value or novelty, please reconsider.

Lines 33-34: no scientific novelty or value

I noticed information about MACE were obtained by telephone interview. How often were the patients reevaluated for scoring system components during the observation period? For instance, lipid profile: how often was it determined and when was the score applied? Were the scores applied more than once after the PCI? How many months after the PCI? Did the scores improve as time went by, or the opposite?

Please elaborate on the relationship between physical activity and MACE. Also, make recommendations on what type of PA is recommended and what type is not in PCI patients and why.

Discussion: “In other words, these two CVH metrics will be maintained and remain unaltered during follow-up, irrespective of whether they develop MACEs or not.” This sentence is unclear, please explain.

Lipids and BMI do not seem to correlate to cardiac adverse events and the relationship between blood pressure and cardiac adverse events is not clearly explained.

Discussion: please discuss any other previously published studies reffering to this topic and their findings.

References: 30 references is a small number, that can be increased by including other previously published papers on this topic.

Author Response

Dear Reviewer:

Thank you for your giving us an opportunity to revise our manuscript! We would like to appreciate the editors and reviewers for their advice and review work to improve the quality of the manuscript. The authors have read each suggestion carefully and found it to be of great help to the article.

Reviewer 2 Report

This is a single-center study with a short follow-up period and a small sample size evaluating the predictive value of 2 cardiovascular health metrics in predicting major cardiac adverse events 2 years after percutaneous coronary intervention for acute coronary syndrome. The article is concisely and clearly written overall but could improve with some minor changes including clarifications in the methods as well as proofreading for grammatical style and word choice.

Please find some detailed feedback below:

- Title: Avoid abbreviations in the title that may not be recognized by everyone; spell out CHS in the title.

- General grammar: incorrect use of words such as "meanwhile" and "herein" throughout the article.

- Methods: Information about MACEs was collected via phone interviews without confirmation with the medical record. Please clarify if there was a way to confirm that what was self-reported by the subjects was correct, in terms of diagnosis, timeline, and severity. If there was no validation performed, it would be important to acknowledge that this may have led to bias or missing data. It should also be clarified at what time point the components of LS7 and LE8 were measured in relation to the timing of the ACS and PCI, as this would have affected the results. It is also unclear how the different components of the LS7 and LE8 were collected - retrospectively or prospectively.

- Page 3 Line 42-44: Sentence starting with "then" needs to be reworded, difficult to understand

- Page 4 Line 6-8: difficult to understand what the LS7 score is for MACE and non-MACE groups, please clarify

- Should acknowledge in the limitations that the patient sample originated from a homogenous population (same country, presumably similar culture/eating habits/lifestyle) and results may not be generalizable to other countries/cultures

Author Response

(The authors gave the same response as above.)

Reviewer 3 Report

I would like to thank Gao et al for the opportunity to review the manuscript of their interesting article "Predictive value of CHS for health outcomes in patients with PCI: Comparison between Life's Simple 7 and Life's Essential 8". In this article, the authors showed that the assessment of CHS in patients with ACS using these scales makes it possible to predict the development of MACE over the next two years. In addition, a comparison of the two scales showed the advantage of the Life's Essential 8 scale in predicting MACE. The results of this study can be used both by researchers on this issue and by practitioners in the development of individual programs for secondary prevention in patients after PCI. However, when reviewing the manuscript, I had comments and questions that I would like to receive answers from the authors.

1.      In the Introduction, the authors assert "These indicate that maintaining ideal CVH may play a crucial role in preventing future MACEs in PCI patients (9)". This statement is inaccurate, since the study Ahmad et al. did not include patients after PCI, but individuals without a history of coronary artery disease.

2.      Among the revascularization procedures in prospective observation, the authors indicate PCI and percutaneous balloon dilatation. Were there indications for coronary artery bypass grafting in the examined cohort of patients? Or is such a revascularization procedure not performed in this clinic?

3.      In the Results section, the authors note: "Compared with non-MACEs groups, the LS7 score was significantly lower in the MACEs group compared to the non-MACEs groups (59.42 ± 11.21, P < 0.001) in the MACEs group (Table 1)". At the same time, the maximum values on the LS7 scale cannot exceed 14 points.

4.      Last row in table 1: "LE8† 55.88 ± 12.12 59.42 ± 11.21 59.64 ± 11.19 <0.001" causes confusion. It turns out that the values of the LE8 scale in the whole cohort are lower than the values of this scale in the two compared groups.

5.      The authors also state that "According to the ROC curve, it can be deduced that LS7 and LE8 had an excellent discriminative performance for differentiating MACEs, with AUCs of 0.615 and 0.662, respectively". It is difficult to agree with this, since in both cases the areas under the curves were < 0.7, indicating insufficient discrimination ability. There is no doubt that the incidence of MACE in this category of patients is influenced by clinical factors (the severity of coronary artery disease, the extent of myocardial infarction, the severity of heart failure). And only taking into account these factors in a complex predictive model will help to achieve excellent discrimination.

Author Response

(The authors gave the same response as above.)

Round 2

Reviewer 1 Report

The authors have properly adressed my comments. Thank you!

Author Response

Dear Reviewer:

We thank you for approving our revision this round. Their valuable comments are very important for the revision and improvement of this manuscript.

Response to Reviewer 1 Comments

We thank Reviewer 1 for the valuable and constructive comments.

Point 1: The authors have properly adressed my comments. Thank you!

Response 1: We thank Reviewer 1 for approving our revision this round. Their valuable comments are very important for the revision and improvement of this manuscript.

Reviewer 3 Report

Thanks to the authors for the detailed answers to my comments and questions. Regarding the answer to my second question. If patients referred for CABG were not included in this population, then this fact should be added to the study limitations.

Author Response

Dear Reviewer:

Thank you for your giving us an opportunity to revise our manuscript in this round! We would like to appreciate the reviewers for their advice and review work to improve the quality of the manuscript. The authors have read each suggestion carefully and found it to be of great help to the article.

Response to Reviewer 3 Comments

We thank Reviewer 3 for the valuable and constructive comments.

Point 1: Regarding the answer to my second question. If patients referred for CABG were not included in this population, then this fact should be added to the study limitations.

Response 1: Thank you very much for your valuable comments. We have added the limitations of this study. Specific changes are as follows: " Besides, our center mainly performs PCI treatment, CABG patients were not included in this sample.  "(Page 10 Line 6-7)